

# Evaluation of flushing time, groundwater discharge and associated nutrient fluxes in Daya Bay, China

Yan Zhang[1,2,3†], Meng Zhang[4†], Hailong Li[2,3*], Xuejing Wang[3], Wenjing Qu[1,2], Xin Luo[5,6], Kai Xiao[1,2], Xiaolang Zhang[3]

[1]MOE Key Laboratory of Groundwater Circulation and Environment Evolution and School of Water Resources and Environment, China University of Geosciences-Beijing, Beijing 100083, China

[2]State Key Laboratory of Biogeology and Environmental Geology, China University of Geosciences-Beijing, Beijing 100083, China

[3]School of Environmental Science and Engineering and Shenzhen Key Laboratory of Soil and Groundwater Pollution Control, Southern University of Science and Technology, Shenzhen 518055, China

[4]Zhong Di Bao Lian (Beijing) Land and Resource Exploration Technology Co., Ltd, Beijing 100193, China

[5]Department of Earth Sciences, The University of Hong Kong, Hong Kong, China

[6]Shenzhen Research Institude, The University of Hong Kong, Shenzhen, China

[†]These authors contributed equally to this work.

* Corresponding author: H. Li (E-mail: hailongli@cugb.edu.cn), Corresponding address: School of Water Resources and Environment, China University of Geosciences-Beijing, Beijing 100083, P. R. China.



## Abstract

Radium quartet have been widely used to quantify the flushing time of water body and submarine groundwater discharge (SGD) in coastal zones. However, previous apparent age model based on mass balance of radium isotopes usually neglected the effects of rivers, open sea water end-member, sedimentary input, atmospheric deposits and recirculated seawater (RSGD). To enhance accuracy in estimating flushing time and SGD, here we present an improved model and then apply in Daya Bay, China. The flushing time estimated by the improved model is 11.8–27.7 d in Daya Bay. It is found that the previous model overestimated the flushing time by 10.7 %–103 %. Considering the radium losses caused by RSGD, the SGD flux is estimated to be $(3.87–5.09)\times10^7 \, m^3 \, d^{-1}$ based on the derived flushing time. The SGD associated nutrient fluxes are estimated to be $(1.36–1.76)\times10^6 \, mol \, d^{-1}$ and $(2.53–3.26)\times10^4 \, mol \, d^{-1}$ for DIN and DIP, respectively, about 20 times greater than those from local rivers. The primary production supported by all the external DIN inputs is determined to be 323–390 mg C $m^{-2} \, d^{-1}$, in which SGD provide approximately 73.1 % of total primary production. Our results reveals that SGD plays an important role in nutrient balance and may be responsible for the frequent outburst of red tides in Daya Bay. The present study provides baseline data for evaluating environmental effects in Daya Bay and similar coastal bay systems elsewhere.

**Keywords:** Radium quartet; Flushing time; Submarine groundwater discharge (SGD); Nutrient fluxes; Primary production; Daya Bay



## 1. Introduction

Submarine groundwater discharge (SGD) is any and all flow of water on continental margins from the seabed to the coastal ocean, regardless of fluid composition or driving force (Burnett and Dulaiova, 2003). SGD, driven by both terrestrial and marine forcing components, comprises submarine fresh groundwater discharge (SFGD) and re-circulated saline groundwater discharge (RSGD). Numerous studies have shown that SGD is an important pathway for the inputs of fresh groundwater, dissolved nutrients, trace metals, carbon and natural radionuclides from coastal aquifers to the ocean (Li et al., 1999; Rosellas et al., 2015; Lecher et al., 2015). These SGD associated chemicals may lead to serious environmental problems such as red tides (Hu et al., 2006; Lee and Kim, 2007; Lee et al., 2010) and benthic macro-algal eutrophication (Hwang et al., 2005). Thus, SGD represents an important process of land/ocean interactions in the coastal ecosystems.

As a major component of global water cycle, SGD has been quantified using different methods such as seepage meters (e.g., Michael et al., 2003; Taniguchi et al., 2005), numerical simulations (Li et al., 2008; Heiss et al., 2014), generalized Darcy's law based on field salinity and groundwater head measurements (Ma et al., 2015; Hou et al., 2016; Qu et al., 2017) and geochemistry tracing method (Moore, 1996; Moore et al., 2008). Naturally occurring radium and radon isotopes are effective tracers for the evaluation of SGD from small to large scales (Paytan et al., 2015; Rodellas et al., 2015; Zhang et al., 2016). Radium has four commonly used isotopes (radium quartet), $^{223}$Ra, $^{224}$Ra, $^{226}$Ra and $^{228}$Ra, with half-lives of 11.4 days, 3.66 days, 1600 years and 5.7 years, respectively. Radium isotopes are produced in situ continuously via decay of their parent thorium isotopes in sediments. They are highly concentrated in coastal groundwater due to high desorption rate and mobility in saline environment (Krishnaswami et al., 1982; Luo et al., 2000).

Daya Bay is a semi-enclosed bay along the coast of Guangdong Province, China. Daya Bay and the surrounding area have been listed as a key economic development zone in China (Arbi et al., 2017). With the rapid economic growth and urban development, this area has been occupied by various kinds of industries, shellfisheries and nuclear power plants during the recent decades (Song et al., 2009). Excess nutrients discharged into the bay due to natural or anthropogenic inputs of point and non-point sources (Wang et al., 2008). As a result, harmful algal blooms have occurred almost every year (especially in spring and summer) in Daya Bay since the 1980s, leading to the deterioration of ecosystem in this area



(Song et al., 2009; Wu et al., 2009; Wang et al., 2014). Wang et al. (2018) indicated that nutrient flux from SGD is the
primary source among all the external inputs of nutrients in Daya Bay and it plays an important role in nutrient structure.
However, the primary production supported by nutrient inputs via SGD and its environmental effect is seldom reported.
The objectives of this study are to estimate the SFGD, flushing time, SGD and associated nutrient fluxes, and then
evaluate the potential effects of the nutrient fluxes on ecological environments in Daya Bay. To improve the accuracy of
these results, mass-balance models of the radium quartet are proposed to consider all the possible source and sink terms.
Uncertainty analyses of the flushing time and SGD with respect to radium activities of groundwater and nearshore
seawater are performed. The nutrient inputs through SGD are compared with those from the local rivers and other external
sources. The potential environmental impacts of the SGD associated nutrient fluxes and the primary production supported
by SGD are evaluated.
**2. Materials and Methods**
**2.1 Study area**
Daya Bay (22.48° N–22.87° N, 114.44° E–114.86° E) is located in the eastern coastline of Guangdong Province, China
(Fig. 1). It is a semi-enclosed shallow bay and approximately covers a water area of ~556 km$^2$. Daya Bay is composed of a
series of sub-basins, including Yaling Bay and Aotou Harbor in the northwest, Fanhe Harbor in the northeast and Dapeng
Cove in the southwest. The tortuous coastline of the bay is about 165 km long. The water depth ranges from 4.5 to 20.4 m
with an average of 11 m. The seafloor sediments are mainly composed of clay and silt (Li et al., 2002). The climate in the
region is subtropical monsoon climate and influenced by the Pacific Ocean. The annual precipitation is 1500–2000 mm,
mainly from May to September (Lin et al., 2002; Zhang et al., 2004). The tide regime is dominated by irregular
semidiurnal tides with an average tidal range of 1 m (Li et al., 1993; Lin et al., 2002; Wang et al., 2014). No large rivers
flow into Daya Bay, but there are four small rivers (Danao River, Huangsuzhou River, Zhuyuan River and Baiyun River)
that discharge into the north coastal part of the bay. Among them the Danao River is the largest one with a discharge rate of
2.62–7.18 m$^3$ s$^{-1}$ (Ren et al., 2013).



## 2.2 Field sampling

A field campaign was conducted to collect seawater, groundwater and river water samples from 28 to 31 July 2015 in Daya Bay. The sampling stations were designed and shown in Fig. 1. For the nearshore stations where the water depth is less than 5 m, only surface seawater (0.5–1.5 m below the surface) was collected. Both surface and bottom (0.5–1.5 m above the seabed) layers were sampled for stations where the seawater depth is more than 5 m. The detailed information about the seawater depth and sampling depth at each station is presented in Table S1. Coastal groundwater samples in the intertidal zone were pumped with a push-point sampler at depths of 0.5–1.5 m. For river water, 30 L of water was collected from ~0.5 m depth below the water surface. These seawater, groundwater and river water samples were filtered through a 0.45 µm polypropylene cartridge filter. Then radium was extracted from the filtered water by passing through a column filled with ~25 g of manganese coated acrylic fiber (Mn-fibers) with a flow rate less than 1 L min$^{-1}$ (Moore et al., 2008). Groundwater with high activity of radium from station GW3 was passed through two columns of Mn-fibers in series to check the extraction efficiency. The environmental indicators such as salinity, temperature, dissolved oxygen (DO) and pH were measured immediately upon the samples collection using a handheld HANNA multi-probe. In addition, nutrient samples for seawater, river water and groundwater were collected in 50 ml plastic vials in the field. After these samples were filtered, they were preserved with saturated $HNO_3$ solution and stored at ~4 °C in the freezer before analysis.

## 2.3 Chemical analysis

The activities of $^{223}Ra$, $^{224}Ra$, $^{226}Ra$ and $^{228}Ra$ were measured by Radium Delayed Coincidence Counter (RaDeCC) (Moore and Arnold, 1996). $^{224}Ra$ was measured within 1–4 days after sample collection to avoid significant decay. $^{223}Ra$ was determined within 7–9 days to avoid significant interference from $^{224}Ra$. The second $^{224}Ra$ measurement was performed approximately one month after sampling to estimate the $^{228}Th$ activity in water. The detecting uncertainties of $^{223}Ra$ and $^{224}Ra$ are 12 % and 7 %, respectively (Moore and Arnold, 1996; Gonneea et al., 2008; Moore and Cai, 2013). After the measurements of $^{223}Ra$ and $^{224}Ra$, each Mn-fiber was stored for 22 days to allow $^{226}Ra$ to equilibrate with $^{222}Rn$ and its daughter nuclides, and then $^{226}Ra$ was measured with RAD7 using the method reported by Kim et al. (2001). $^{228}Ra$ was determined via measuring $^{228}Th$ ingrown from $^{228}Ra$ and calculating the initial $^{228}Ra$. The detecting uncertainties of $^{226}Ra$



and $^{228}$Ra are ~20 % and 7 %, respectively (Moore et al., 2008; Luo and Jiao, 2016). The extraction efficiency was
estimated to be 99.86 %, indicating a well recovery.

The dissolved nitrate ($NO_3$), nitrite ($NO_2$), ammonium ($NH_4$), inorganic nitrogen (DIN, sum of $NO_3$, $NO_2$ and $NH_4$)

and reactive phosphorus (DIP) were analyzed for surface seawater, river water and groundwater with a spectrophotometer
in National Research Centre of Geoanalysis (NRCG). Analytical uncertainties are < 3 % for $NO_3$, < 8 % for $NO_2$, < 10 %
for $NH_4$ and < 5 % for DIP.

## 2.4 Radium mass balance models

Radium mass balance models have been widely used to quantify SGD in the coastal areas. However, these models
neglected the losses of radium in seawater caused by RSGD. Zhang et al. (2017) reported that neglecting the losses of
tracers induced by RSGD would underestimate the SGD and they presented an improved mass balance model which
considers the radium losses that RSGD takes away from seawater to assess the SGD in Jiaozhou Bay, China. Here we
made similar attempt and applied the improved model to Daya Bay. For this bay system, sources of radium mainly include
SGD, riverine input, atmospheric deposit, rainfall input and sedimentary input. The sinks mainly include radioactive decay,
mixing loss and the losses of radium in seawater caused by RSGD. The input fluxes from precipitation can be negligible
(Moore et al., 2008). Under steady state, the radium inputs into the system should be balanced by outputs. Thus, the
radium mass balance model can be expressed as follows (Zhang et al., 2017):
$$Q_{SGD}\,^{22n}Ra_{gw} +\,^{22n}F_r +\,^{22n}F_{atm} +\,^{22n}F_{sed} = \frac{I_{22n} - V\,^{22n}Ra_{op}}{T_f} + I_{22n}\lambda_{22n} + Q_{RSGD}\,^{22n}Ra_{ns} \tag{1a}$$

where $Q_{SGD}$ and $Q_{RSGD}$ are the SGD and RSGD fluxes, respectively; $n$ equals 3, 4, 6 and 8 for representing $^{223}$Ra,
$^{224}$Ra, $^{226}$Ra and $^{228}$Ra, respectively; $^{22n}Ra_{gw}$, $^{22n}Ra_{op}$ and $^{22n}Ra_{ns}$ are the end-members (i.e., representative activity
values) of $^{22n}Ra$ in groundwater, open sea water and nearshore seawater, respectively; $^{22n}F_r$, $^{22n}F_{atm}$ and $^{22n}F_{sed}$
are the $^{22n}Ra$ inputs from rivers, atmospheric deposits and sediments; $I_{22n}$ is the inventory of $^{22n}Ra$ in the bay; $V$ is





the seawater volume of the bay; $T_f$ is the flushing time; $\lambda_{22n}$ is the decay constant of $^{22n}Ra$. The term
$Q_{RSGD}\,^{22n}Ra_{ns}$ represents the losses of radium that RSGD takes away from seawater when it invades coastal aquifers.
The RSGD flux can be obtained by subtracting SFGD from SGD, i.e.,
$$Q_{RSGD} = (1 - R_F)Q_{SGD} \qquad\qquad (1b)$$
where $R_F$ is the ratio of SFGD flux to SGD flux.
**3. Results**
**3.1 Environmental factors**
Results of environmental parameters (salinity, pH and temperature) for seawater are shown in Supplementary Table S1 and
Fig. S1. In general, the salinity of surface and bottom layers increases from northeast to the mouth of the bay. Surface
salinity exhibits significant variations among stations (16.2–33.3) and low values are observed in the nearshore areas and
estuaries, indicating the effects of fresh groundwater and river water. The bottom seawater shows higher salinity with a
value of 28.6–32.5 during the sampling period. Seawater pH has a relatively narrow range throughout the bay, varying
from 7.74 to 8.75 in surface layer and 7.98 to 8.33 in bottom layer. Temperature in surface and bottom seawater tends to
decrease from nearshore to offshore waters. The mean pH and temperature for surface seawater are slightly higher than
those in bottom layer.
**3.2 Radium isotopes**
The activities of $^{223}$Ra, $^{224}$Ra, $^{226}$Ra and $^{228}$Ra in seawater are listed in Supplementary Table S1. The radium activities for
surface seawater exhibit noticeable spatial differences, varying from 1.09 to 8.87 dpm 100 L$^{-1}$ for $^{223}$Ra, 15.1 to 294 dpm
100 L$^{-1}$ for $^{224}$Ra, 8.29 to 31.3 dpm 100 L$^{-1}$ for $^{226}$Ra and 13.7 to 100 dpm 100 L$^{-1}$ for $^{228}$Ra. The activities of bottom
seawater ranges from 1.06 to 7.06 dpm 100 L$^{-1}$ for $^{223}$Ra, 25.7 to 119.4 dpm 100 L$^{-1}$ for $^{224}$Ra, 5.92 to 31.6 dpm 100 L$^{-1}$ for
$^{226}$Ra and 1.49 to 80.0 dpm 100 L$^{-1}$ for $^{228}$Ra.



As shown in Fig. 2, one can see that the four radium isotopes in surface seawater share the following distribution features: (1) high activities are detected in the nearshore areas and estuaries, and lowest activities are observed in the mouth of the bay; and (2) activities generally decrease from northeast to southwest of the bay. Only surface seawater at the seven nearshore stations (S2–S3, S7, S15–S17, S19) was collected due to the shallow bay water. If excluding these nearshore stations, radium activities in bottom layer show similar distribution pattern as the surface water. Moreover, the mean radium activities of surface and bottom layers are found to be close to each other and certainly consistent within their respective measurement errors, indicating that the water column is well mixed.

### 3.3 Nutrients

The concentrations of nutrients for 13 surface seawater samples, 9 coastal groundwater samples and 3 river water samples are shown in Supplementary Table S2. The concentrations of nutrients in surface seawater exhibit spatial differences and range from 1.43 to 92.14 µmol L$^{-1}$ for NO$_3$, 0.09 to 43.48 µmol L$^{-1}$ for NO$_2$, 0.09 to 135.62 µmol L$^{-1}$ for DIN and 0.20 to 2.98 µmol L$^{-1}$ for DIP. In general, a decreased trend is observed in nutrient concentrations from the northeast part to the mouth of the bay. For the groundwater samples from sandy beach aquifers, the mean DIN (88.3 µmol L$^{-1}$) and DIP (1.73 µmol L$^{-1}$) concentrations are ~2 times higher than those in surface seawater. In this study area, only one groundwater sample (station GW2) has high NH$_4$ content (97.78 µmol L$^{-1}$), and no NH$_4$ is detected in surface seawater, river water and other groundwater samples. Generally, NO$_3$ is the important component of DIN, making up about 77.3 % and 68.1 % of DIN for groundwater and surface seawater, respectively. Qu et al. (2017) also reported the absence phenomenon of NH$_4$ in coastal groundwater at a sandy beach of Jiaozhou Bay, China.

## 4. Discussion

### 4.1 Water mass balance model

Drainage basin water mass balance model is used to estimate the SFGD flux in Daya Bay. In Daya Bay basin, the main water input is precipitation. The losses include evapotranspiration, river discharge and SFGD. If the change in storage of groundwater in the basin is ignored (steady-state), water mass balance can be written as follows:



$$Q_{SFGD} = P_T - E_T - \sum_{i=1}^{m} Q_{r(i)} \qquad (2)$$

where $Q_{SFGD}$ is the SFGD flux; $P_T$ is the precipitation; $E_T$ is the evapotranspiration; $m=4$ is the number of major
rivers discharging into Daya Bay; $Q_{r(i)}$ is the discharge of the $i$-th river. The area of Daya Bay basin is about 870 km².
The annual mean precipitation is 1800 mm, which is the average value during the 11 years from 2005 to 2015 obtained
from National Meteorological Information Center (NMIC). The annual mean evapotranspiration of 788 mm is estimated
by the Gaoqiao equation using air temperature and precipitation data from 2005 to 2015. With the area of the basin and
daily precipitation and evapotranspiration, $P_T$ and $E_T$ can be estimated to be $4.57 \times 10^6$ m³ d$^{-1}$ and $2.01 \times 10^6$ m³ d$^{-1}$,
respectively. The discharge of all the rivers around Daya Bay is $6.75 \times 10^5$ m³ d$^{-1}$ (Table 1). Substituting all the obtained
values into Eq. (2), the SFGD flux can be estimated to be $1.89 \times 10^6$ m³ d$^{-1}$ or $3.40 \times 10^{-3}$ m d$^{-1}$ if divided by the area (556
km²) of Daya Bay.

SFGD in different coastal systems from small to large scales has been assessed in several previous studies. Lee et al.

(2012) used water mass balance model to assess a SFGD flux of $(4–11) \times 10^{-3}$ m d$^{-1}$ in Tolo Harbour of Hong Kong with an
area of 52 km². Crotwell and Moore (2003) employed salinity mass balance model to estimate the SFGD to be $(9–14) \times 10^{-3}$
m d$^{-1}$ in Port Royal Sound, South Carolina with an area of 300 km². Wang et al. (2015) coupled water and salt mass
balance models and obtained a SFGD flux of $(7–10) \times 10^{-3}$ m d$^{-1}$ in Laizhou Bay, China which has an area of 6000 km². Liu
et al. (2017) employed two-end-member model of radium isotopes and evaluated the SFGD to be $(1.78–5.34) \times 10^{-4}$ m d$^{-1}$ in
the Bohai Sea, China which essentially has large water area of 77000 km². When compared our SFGD flux in Daya Bay
with their estimated SFGD in different scales, our estimate falls in the ranges of the results reported by previous studies.
**4.2 Tidal prism model**
To explore the dynamics of coastal water in Daya Bay, the flushing time related to the timescale of material transport is
estimated. Flushing time $T_f$ is an important physical parameter for describing the general exchange characteristics of the





water body in a bay system (Monsen et al., 2002). It is the ratio of the mass or volume of a constituent to its renewal rate.
One can estimate the flushing time based on the tidal prism method (Sanford et al., 1992):
$$T_f = \frac{VT}{P(1-b)}$$
(3a)

$$P = \int_0^H S \, dz \approx HS$$
(3b)

where $T$ is the tidal period with a value of 0.5 d for the typical semidiurnal tides of the study area; $P$ is the tidal prism,
i.e., total volume of seawater entering the bay during a rising tide; $b$ is the return flow factor (percentage of the tidal prism
that returns from outside of the bay during the flood tide); $S$ is the water surface area of the bay that varies with water
depth $z$, which ranges from low tide (elevation datum) to high tide; and $H$ is the tidal range. Because the intertidal zone is
narrow (Li et al., 1993; Lin et al., 2002; Wang et al., 2014), the change of $S$ over the tidal cycle can be neglected. Thus,
the tidal prism can be simply approximated by $HS$. The mean seawater volume $V$ of the bay is $4.39 \times 10^9$ m$^3$ and the
calculation details are given in section 4.3.1. During the sampling period, the average area $S$ of the bay is $5.56 \times 10^8$ m$^2$ and
the tidal range $H$ is 1.08 m. Thus the tidal prism $HS$ equals $6.00 \times 10^8$ m$^3$. In Eq. (3a), the return flow factor is defined as (Van
de Kreeke, 1983):
$$b = \frac{v-U}{v+U}$$
(3c)

where $v$ is the average value of the flood and ebb velocity; $U$ is the net velocity (the difference between incoming flood
velocity and the outflowing ebb velocity). The average velocity of rising tide and falling tide is 30.8 cm s$^{-1}$ and 34.3 cm s$^{-1}$,
respectively (Ma et al., 1998). Based on Eq. (3c), one can obtain a return flow factor of 0.81. Substituting the parameters
above into Eq. (3a), the flushing time is estimated to be 18.8 d.
**4.3 Improved flushing time model**
Moore (2000) and Moore et al. (2006) developed the apparent age model based on mass balance of radium quartet, which
has been used to quantify the flushing time in many previous studies (e.g., Peterson et al., 2008; Ji et al., 2013;
Tomasky-Holmes et al., 2013; Xu et al., 2013; Wang et al., 2015; Luo and Jiao, 2016). However, the model assumed that



groundwater is the major source, and neglected other sources such as riverine input, sedimentary input and atmospheric
deposits. Moreover, it did not consider the effects of open sea water end-member and RSGD. In this case, estimates of
flushing time would be of considerable uncertainties. Here we take into account new sources (radium inputs from rivers,
sediments and atmospheric deposits) and sink term (the loss caused by RSGD) to enhance the accuracy of flushing time by
coupling two radium mass balance models.
Substituting Eq. (1b) into (1a) and rearranging, yield
$$Q_{SGD}[{}^{22n}Ra_{gw}-(1-R_F){}^{22n}Ra_{ns}]+{}^{22n}F_r+{}^{22n}F_{sed}+{}^{22n}F_{atm}=\frac{I_{22n}-V{}^{22n}Ra_{op}}{T_f}+I_{22n}\lambda_{22n} \tag{4}$$

Dividing the two equations of (4) corresponding to ${}^{22j}$Ra and ${}^{22i}$Ra to eliminate the term $Q_{SGD}$, and then solving for
$T_f$, one has
$$T_f=\frac{{}^{22j/22i}Ra_{gw\text{-}ns}(I_{22i}-V{}^{22i}Ra_{op})-(I_{22j}-V{}^{22j}Ra_{op})}{(\lambda_{22j}I_{22j}-{}^{22j}F_r-{}^{22j}F_{sed}-{}^{22j}F_{atm})-{}^{22j/22i}Ra_{gw\text{-}ns}(\lambda_{22i}I_{22i}-{}^{22i}F_r-{}^{22i}F_{sed}-{}^{22i}F_{atm})} \tag{5a}$$

where the term ${}^{22j/22i}Ra_{gw-ns}$ is defined as
$${}^{22j/22i}Ra_{gw\text{-}ns}=\frac{{}^{22j}Ra_{gw}-(1-R_F){}^{22j}Ra_{ns}}{{}^{22i}Ra_{gw}-(1-R_F){}^{22i}Ra_{ns}} \tag{5b}$$

and $j$ equals 3 or 4; $i$ equals 6 or 8. The term ${}^{22j/22i}Ra_{gw-ns}$ will be referred to as the 22j/22i ratio of activity difference
between groundwater and nearshore seawater hereafter.
**4.3.1 Water volume and radium inventory in Daya Bay**
In order to calculate the seawater volume and radium inventory of Daya Bay, the bay is divided into 93 triangle elements
(Fig. S2). The water volume is calculated as the sum of the volume in the seawater column corresponding to each small
triangle element. The volume of each triangular prism is calculated as the product of the area of triangle element and average
water depth at the three vertexes. Based on the measured coordinates and water depths, the area and volume in the bay is
calculated to be $5.56\times10^8$ m$^2$ and $4.39\times10^9$ m$^3$, respectively. The radium inventory is calculated as the sum of the inventory in





the seawater column corresponding to each triangle element. The inventory for each seawater column is calculated as the
product of the volume of seawater and the average value of the radium activity at the three vertexes of the triangle element.
The entire bay water is considered to be vertically well mixed, thus radium activities in surface and bottom layers (if any) at
each station are averaged to represent the activity values at the station. The total radium inventories are calculated to be
$1.36\times10^{11}$ dpm for $^{223}$Ra, $2.59\times10^{12}$ dpm for $^{224}$Ra, $6.81\times10^{11}$ dpm for $^{226}$Ra and $1.42\times10^{12}$ dpm for $^{228}$Ra.
**4.3.2 Radium inputs from rivers and atmospheric deposits**
Riverine inputs include the dissolved radium in river water and desorption from suspended particulate matter (SPM) in river
water, which can be written as follows:
$$^{22n}F_r = \sum_{i=1}^{m}\left(Q_{r(i)}\,^{22n}Ra_{r(i)} + {}^{22n}SPM_{r(i)}\right) \tag{6}$$

where $^{22n}Ra_{r(i)}$ is the activity of radium in the $i$-th river; $^{22n}SPM_{r(i)}$ is the desorption flux from SPM in the $i$-th river
water.
Using the discharges and radium activities in each river as shown in Table 1, the dissolved flux is estimated to be
$1.24\times10^{7}$ dpm d$^{-1}$ for $^{223}$Ra, $3.92\times10^{8}$ dpm d$^{-1}$ for $^{224}$Ra, $1.63\times10^{8}$ dpm d$^{-1}$ for $^{226}$Ra and $1.06\times10^{8}$ dpm d$^{-1}$ for $^{228}$Ra.
Radium quartet have strong adsorption to particulates in freshwater (Luo et al., 2000). Radium adsorbed in SPM could be
released into seawater after river water enters the saline environments and mixes with the seawater. The flux of radium
desorption from SPM is the product of riverine particle flux and desorption rate of radium. Wang et al. (2018) reported that
the total flux of SPM from four rivers is $1.16\times10^{7}$ g d$^{-1}$ in winter 2015 in Daya Bay. Based on their result, the riverine SPM
flux is estimated to be $2.01\times10^{7}$ g d$^{-1}$ in summer. The maximum desorption rate of radium from SPM is 0.1 dpm g$^{-1}$ for
$^{223}$Ra, and 2 dpm g$^{-1}$ for $^{224}$Ra, $^{226}$Ra and $^{228}$Ra (Moore et al., 2011; Luo et al., 2014, 2018; Moore, 1996; Moore et al.,
2008). Thus the total inputs of radium from rivers are $1.45\times10^{7}$ dpm d$^{-1}$ for $^{223}$Ra, $4.32\times10^{8}$ dpm d$^{-1}$ for $^{224}$Ra, $2.03\times10^{8}$
dpm d$^{-1}$ for $^{226}$Ra and $1.46\times10^{8}$ dpm d$^{-1}$ for $^{228}$Ra.
Assuming that radium desorption from atmospheric deposit is similar to desorption from riverine SPM, atmospheric
deposit of radium could be determined as the product of deposit rate of SPM, desorption rate of radium and water area.





The SPM deposit rate is 0.01 g m$^{-2}$ d$^{-1}$ in summer in Daya Bay (Du et al., 1994). Thus the radium inputs from atmospheric

deposit in the entire bay are estimated to be 7.96×10$^5$ dpm d$^{-1}$ for $^{223}$Ra, and 1.59×10$^7$ dpm d$^{-1}$ for $^{224}$Ra, $^{226}$Ra and $^{228}$Ra,

which are 1–2 orders of magnitude lower than those from the local rivers.

**Table 1.** Discharges, dissolved radium activities and radium fluxes from all rivers.

| River | Discharge (m$^3$ d$^{-1}$) | Radium activity (dpm 100 L$^{-1}$) | | | | Riverine flux (10$^6$ dpm d$^{-1}$) | | | |
|---|---|---|---|---|---|---|---|---|---|
| | | $^{223}$Ra | $^{224}$Ra | $^{226}$Ra | $^{228}$Ra | $^{223}$Ra | $^{224}$Ra | $^{226}$Ra | $^{228}$Ra |
| R1 | 4.71×10$^5$ | 2.07 | 73.8 | 26.5 | 18.0 | 9.89 | 375 | 153 | 113 |
| R2 | 5.4×10$^4$ | 1.47 | 34.2 | 21.0 | 15.7 | 0.81 | 21.7 | 14.6 | 11.7 |
| R3 | 4.23×10$^4$ | 0.72 | 13.4 | 10.5 | 4.89 | 0.32 | 8.19 | 6.98 | 4.59 |
| R4 | 1.08×10$^5$ | 1.48 | 18.6 | 20.5 | 10.3 | 1.63 | 26.5 | 28.6 | 17.6 |
| Total flux | 6.75×10$^5$ | | | | | 12.6 | 432 | 203 | 146 |

### 4.3.3 Radium input from sediments

Radium input from sediments is an important source for short-lived isotopes, which includes diffusion, physical mixing,

bioturbation and bioirrigation (Moore et al., 2011; Luo and Jiao, 2016). Although we did not directly carry out sediment

leaching experiments, the radium input from sediments could be estimated using the following formulation (Moore et al.,

2011):

$$^{22n}F_{sed} = S\,^{22n}Pn\sqrt{\frac{(D_t + D_{mix}K_{22n})\lambda_{22n}}{1 + K_{22n}}} \tag{7}$$

where $^{22n}P$ is the production rate of $^{22n}Ra$ in sediments; $n$ is the porosity of sediments; $K_{22n}$ is the adsorption

coefficient of $^{22n}Ra$; $D_t$ and $D_{mix}$ are the dispersion coefficient and sediment mixing coefficient, respectively. $D_t$

and $D_{mix}$ have the values of 1.73×10$^{-4}$ m$^2$ d$^{-1}$ and 5.18×10$^{-5}$ m$^2$ d$^{-1}$ for coastal sediments, respectively (Moore et al.,

2011).





The contents of uranium (U) and thorium (Th) in coastal sediments of Guangdong Province, China are 2.06 ug g$^{-1}$ and
10.4 ug g$^{-1}$, respectively (Li and Liu, 1987). One can derive the $^{232}$Th, $^{238}$U and $^{235}$U activities in sediments to be 2.60 dpm
g$^{-1}$, 1.58 dpm g$^{-1}$ and 0.07 dpm g$^{-1}$, respectively based on the U and Th contents (Kraemer, 2005). When parent and
daughter nuclides for $^{235}$U-series, $^{238}$U-series and $^{232}$Th-series approach secular equilibrium, the activities of radium quartet
can be obtained to be 0.07 dpm g$^{-1}$ for $^{223}$Ra, 2.60 dpm g$^{-1}$ for $^{224}$Ra, 1.58 dpm g$^{-1}$ for $^{226}$Ra and 2.60 dpm g$^{-1}$ for $^{228}$Ra. The
leachable ratios for $^{223}$Ra, $^{224}$Ra, $^{226}$Ra and $^{228}$Ra are 7 %, 7 %, 0.6 % and 5 %, respectively (Moore et al., 2011). The
sediment density and porosity estimated from soil analysis is 0.88 kg L$^{-1}$ and 0.33, respectively in Daya Bay. Using the
radium activities in sediments, leachable ratios and sediment density, the production rate $^{22n}P$ of radium quartet can be
estimated to be 4.31 dpm, 160 dpm, 8.35 dpm and 114 dpm per liter wet sediment for $^{223}$Ra, $^{224}$Ra, $^{226}$Ra and $^{228}$Ra,
respectively. The median activity of radium in the coastal porewater is 35.1 dpm 100 L$^{-1}$ for $^{223}$Ra, 1078 dpm 100 L$^{-1}$ for
$^{224}$Ra, 38.7 dpm 100 L$^{-1}$ for $^{226}$Ra and 103 dpm 100 L$^{-1}$ for $^{228}$Ra. Dividing the production rate with activity of porewater,
the adsorption coefficients $K_{22n}$ for $^{223}$Ra, $^{224}$Ra, $^{226}$Ra and $^{228}$Ra are estimated to be 12.3, 14.9, 21.5 and 111,
respectively. Substituting all the yield terms into Eq. (7), the inputs from sediments for $^{223}$Ra, $^{224}$Ra, $^{226}$Ra and $^{228}$Ra are
$1.52 \times 10^9$ dpm d$^{-1}$, $9.86 \times 10^{10}$ dpm d$^{-1}$, $1.27 \times 10^7$ dpm d$^{-1}$ and $2.76 \times 10^9$ dpm d$^{-1}$, respectively; or 2.74 dpm m$^{-2}$ d$^{-1}$, 177 dpm
m$^{-2}$ d$^{-1}$, 0.02 dpm m$^{-2}$ d$^{-1}$ and 4.97 dpm m$^{-2}$ d$^{-1}$, respectively (normalized to per square meter of area). These values are
within the ranges of the results derived from previous studies (Hancock et al., 2000, 2006; Crotwell and Moore, 2003;
Moore et al., 2006).
**4.3.4 Flushing time estimation for seven different cases**
Here we use four combinations of radium quartet mass balance models ($^{223}$Ra & $^{226}$Ra, $^{223}$Ra & $^{228}$Ra, $^{224}$Ra & $^{226}$Ra and
$^{224}$Ra & $^{228}$Ra) to evaluate the flushing time of Daya Bay. Table S3 presents the statistical summary of the four data sets
($^{22n}Ra_{gw}$, $^{22n}Ra_{ns}$, $^{22j/22i}Ra_{gw-ns}$ and $^{22n}Ra_{gw-ns}$) from the 9 groundwater samples and corresponding nearshore
seawater samples. The mean $R_F$ of 5 % is used to derive the term $^{22j/22i}Ra_{gw-ns}$, which will be proved later.
The medians of the data set $^{22j/22i}Ra_{gw-ns}$ are used for representative values to estimate the flushing time. The





radium activities measured at station S13 are used as the open sea water end-members. Substituting all the obtained values

into Eq. (5a), the flushing time estimated by $^{223/226}Ra_{gw-ns}$, $^{223/228}Ra_{gw-ns}$, $^{224/226}Ra_{gw-ns}$ and $^{224/228}Ra_{gw-ns}$ is

33.1 d, 31.6 d, 16.5 d and 22.5 d, respectively, with an average of 25.9 d.

Compared to previous apparent age model, our model takes into account the effects of five factors, i.e., radium inputs

from rivers, sediments and atmospheric deposits, open sea water end-member and RSGD. In order to understand how the

five different factors affect the flushing time estimation, we analyze the following seven cases listed in Table 2. From the

results of Case 1 and Case 2, one can see that neglecting the riverine inputs of radium results in small increases (0.44 %–

6.04 %) in flushing time, which indicates that flushing time model is less sensitive to the riverine input due to the small

discharge in the study area. Compared to Case 1, Case 3 neglects the radium inputs from sediments and it underestimates

the flushing time by 0.95 %–19.4 %. The results of flushing time for Case 4 are consistent with the results of Case 1. This

is because atmospheric deposit contributes a quite small portion to the mass balance models and represents a minor

influence on flushing time. From Case 5, it can be seen that ignoring the effect of the open sea water end-members results

in significant increases (128 %–443 %) in flushing time. Case 6 do not consider the radium losses induced by RSGD, thus

the results are underestimated by 34.2 %–69.5 %. Obviously, the flushing time model is strongly influenced by the open

sea water end-member and RSGD in Daya Bay. Case 7 represents the result estimated by previous apparent age model

which ignores all the above-mentioned five factors. In this case, the flushing time would be overestimated by 103 % for

$^{223/226}Ra_{gw-ns}$, 58.5 % for $^{223/228}Ra_{gw-ns}$, 83.0 % for $^{224/226}Ra_{gw-ns}$ and 10.7 % for $^{224/228}Ra_{gw-ns}$. Based on the

analyses above, we can conclude that significant errors may be produced when estimating the flushing time if one do not

consider the effects of the five factors. Therefore, we would recommend the improved flushing time model presented in

Eqs. (5a) and (5b) to derive the flushing time.





16 **Table 2.** Comparisons of flushing time estimations for seven different cases for four combinations of the radium quartet.

| Cases | Rivers | Sediments | Atmospheric deposits | Open sea water | RSGD | Flushing time (d) | | | |
|---|---|---|---|---|---|---|---|---|---|
| | | | | | | $^{223}$Ra & $^{226}$Ra | $^{223}$Ra & $^{228}$Ra | $^{224}$Ra & $^{226}$Ra | $^{224}$Ra & $^{228}$Ra |
| Case 1 | √ | √ | √ | √ | √ | 33.1 | 31.6 | 16.5 | 22.5 |
| Case 2 | × | √ | √ | √ | √ | 35.1 | 31.8 | 16.9 | 22.6 |
| Case 3 | √ | × | √ | √ | √ | 27.4 | 31.3 | 13.3 | 20.2 |
| Case 4 | √ | √ | × | √ | √ | 33.2 | 31.6 | 16.5 | 22.6 |
| Case 5 | √ | √ | √ | × | √ | 180 | 76.2 | 83.3 | 51.5 |
| Case 6 | √ | √ | √ | √ | × | 10.1 | 20.8 | 6.06 | 12.3 |
| Case 7 | × | × | × | × | × | 67.1 | 50.1 | 30.2 | 24.9 |

17 "√"=considered; "×"=not considered



### 4.3.5 Uncertainty analyses of flushing time

A number of factors have been suggested as the cause of these high uncertainties in estimating flushing time.

$^{22j/22i}Ra_{gw-ns}$ is an important term of the improved flushing time model and it may have the largest source of uncertainty.

The values of $^{22j/22i}Ra_{gw-ns}$ span a wide range due to the considerable spatial variations of radium activities in 9

groundwater samples and corresponding nearshore seawater samples. The ratios of the maximum to the minimum values

of $^{22j/22i}Ra_{gw-ns}$ are about 107 for $^{223/226}Ra_{gw-ns}$, 67.0 for $^{223/228}Ra_{gw-ns}$, 53.5 for $^{224/226}Ra_{gw-ns}$ and 33.6 for

$^{224/228}Ra_{gw-ns}$. Thus determining a representative $^{22j/22i}Ra_{gw-ns}$ value may be a challenge.

In order to conveniently discuss how flushing time depends on $^{22j/22i}Ra_{gw-ns}$, we define the relative error

$RE(^{22j/22i}Ra_{gw-ns})$ of $^{22j/22i}Ra_{gw-ns}$ with respect to its median $Q_2(^{22j/22i}Ra_{gw-ns})$ as follows

$$RE(^{22j/22i}Ra_{gw-ns}) = \frac{^{22j/22i}Ra_{gw-ns} - Q_2(^{22j/22i}Ra_{gw-ns})}{Q_2(^{22j/22i}Ra_{gw-ns})} \qquad (8)$$

To consider the large uncertainties of the flushing time induced by $^{22j/22i}Ra_{gw-ns}$, the range between first

($Q_1(^{22j/22i}Ra_{gw-ns})$) and third ($Q_3(^{22j/22i}Ra_{gw-ns})$) quartiles of $^{22j/22i}Ra_{gw-ns}$ in Eq. (8) will be used in the following

discussion.

Figure 3 shows the how the flushing time changes with $RE(^{22j/22i}Ra_{gw-ns})$ for four different combinations of

radium quartet in summer of Daya Bay. Based on Eqs. (5a) and (5b), $T_f$ is estimated to be -1.83–91.15 d for

$^{223/226}Ra_{gw-ns}$, 11.8–41.9 d for $^{223/228}Ra_{gw-ns}$, 3.94–33.8 d for $^{224/226}Ra_{gw-ns}$ and 3.87–27.69 d for $^{224/228}Ra_{gw-ns}$.

One can see that the flushing time estimated by $^{223/226}Ra_{gw-ns}$ has the maximum range of variation. To eliminate the

error of flushing time caused by different radium isotopes, the common intersection of the four value ranges (11.8–27.7 d)

determined by Fig. 3 is selected as the most reasonable result of flushing time, which agrees well with the result from tidal





prism model (18.8 d).
**4.4 SGD estimation and uncertainty analyses**

Substituting the flushing time into Eq. (1a) and solving for $Q_{SGD}$, one has

$$Q_{SGD} = \frac{I_{22n} - V\,^{22n}Ra_{op} + T_f(I_{22n}\lambda_{22n} - \,^{22n}F_r - \,^{22n}F_{sed} - \,^{22n}F_{atm})}{T_f\,^{22n}Ra_{gw\text{-}ns}}$$    (9a)
$$^{22n}Ra_{gw\text{-}ns} = \,^{22n}Ra_{gw} - (1 - R_F)\,^{22n}Ra_{ns}$$    (9b)

The radium quartet ($^{223}$Ra, $^{224}$Ra, $^{226}$Ra and $^{228}$Ra) in Eqs. (8a) and (8b) are applied to determine the SGD in Daya Bay.

The median values of $^{22n}Ra_{gw\text{-}ns}$ are 31.5 dpm 100 L$^{-1}$ for $^{223}$Ra, 976 dpm 100 L$^{-1}$ for $^{224}$Ra, 18.2 dpm 100 L$^{-1}$ for $^{226}$Ra
and 66.2 dpm 100 L$^{-1}$ for $^{228}$Ra (Table S3). Using the range of flushing time of 11.8–27.7 d and median values of
$^{22n}Ra_{gw\text{-}ns}$, the SGD flux in Daya Bay is estimated to be (3.14–4.47)×10$^7$ m$^3$ d$^{-1}$, (4.61–5.38)×10$^7$ m$^3$ d$^{-1}$, (3.07–7.40)×10$^7$
m$^3$ d$^{-1}$ and (3.24–8.13)×10$^7$ m$^3$ d$^{-1}$ for the $^{223}$Ra, $^{224}$Ra, $^{226}$Ra and $^{228}$Ra mass balance models, respectively.

For the improved radium mass balance models, $^{22n}Ra_{gw\text{-}ns}$ associated with groundwater and nearshore seawater

end-members may be an important source of uncertainty in estimating SGD. The ratios of maximum to median values of
$^{22n}Ra_{gw\text{-}ns}$ are approximately 20 for $^{223}$Ra and $^{228}$Ra, and 8 for $^{224}$Ra and $^{226}$Ra. Similar to the estimation of flushing time,
the range between the first and third quartiles of the data set of $^{22n}Ra_{gw\text{-}ns}$ shown in Table S3 is used to evaluate the
uncertainty of SGD induced by $^{22n}Ra_{gw\text{-}ns}$. For the sake of convenience of analysis, we define the relative error of
$RE(^{22n}Ra_{gw\text{-}ns})$ of $^{22n}Ra_{gw\text{-}ns}$ with respect to its median as
$$RE(^{22n}Ra_{gw\text{-}ns}) = \frac{^{22n}Ra_{gw-ns} - Q_2(^{22n}Ra_{gw-ns})}{Q_2(^{22n}Ra_{gw-ns})}$$    (10)
where $^{22n}Ra_{gw\text{-}ns}$ ranges from $Q_1(^{22n}Ra_{gw\text{-}ns})$ to $Q_3(^{22n}Ra_{gw\text{-}ns})$. $Q_1(^{22n}Ra_{gw\text{-}ns})$, $Q_2(^{22n}Ra_{gw\text{-}ns})$ and
$Q_3(^{22n}Ra_{gw\text{-}ns})$ are the first, second (median) and third of quartiles of the data set of $^{22n}Ra_{gw\text{-}ns}$, respectively.





Figure 4 shows how the SGD changes with $RE(^{22n}Ra_{gw})$ for $^{223}$Ra, $^{224}$Ra, $^{226}$Ra and $^{228}$Ra mass balance models.
The mean flushing time determined by four ratios of radium quartet is 19.7 d. Using the mean flushing time, the SGD is
estimated to be $(1.61–5.13)\times10^7$ m$^3$ d$^{-1}$, $(2.20–7.27)\times10^7$ m$^3$ d$^{-1}$, $(1.58–5.09)\times10^7$ m$^3$ d$^{-1}$ and $(3.87–6.70)\times10^7$ m$^3$ d$^{-1}$ for the
$^{223}$Ra, $^{224}$Ra, $^{226}$Ra and $^{228}$Ra mass balance models, respectively. Based on their intersection, the range of $(3.87–5.09)\times10^7$
m$^3$ d$^{-1}$ is used as the final SGD result. One can see that the SGD flux exceeds the total discharges of the local rivers by a
factor of 57.3–75.4. The SFGD accounts for only about 4.30 % of SGD and the RSGD is the primary component of SGD
in Daya Bay. The parameters and values used in the radium quartet mass balance models are summarized in Table 3.
In addition to $^{22n}Ra_{gw\text{-}ns}$, there are some other sources of uncertainties such as the decay loss, mixing loss and
radium inputs from sediments and rivers. Both decay and mixing losses are related to radium inventories. The uncertainty
of radium inventories are mainly from the measurement error (12 % for $^{223}$Ra, 20 % for $^{226}$Ra, 7 % for $^{224}$Ra and $^{228}$Ra).
Based on the law of error of propagation, such measurement error will result in a variation in SGD about by 17.1 % for
$^{223}$Ra model, 9.21 % for $^{224}$Ra model, 73.6 % for $^{226}$Ra model and 16.9 % for $^{228}$Ra model. The open sea water end-member
associated with mixing loss is determined by one seawater sample (station S13) collected from outside of the bay. Using
only one sample may not provide representative end-member values. Thus the radium activities measured at station S13
are changed by ± 20 % (the maximum measurement error of radium) to consider the uncertainty induced by the open sea
end-member. In this case, SGD would change by < 5 % for $^{223}$Ra and $^{224}$Ra models, 70.5 % for $^{226}$Ra model and 26.1 % for
$^{228}$Ra model. The sedimentary input is an important source for short-lived radium isotopes and it is equivalent to 10.1 %
and 16.1 % of total inputs for $^{223}$Ra and $^{224}$Ra mass balance models, respectively. If we have assigned an uncertainty of
20 % in the sedimentary input, the SGD would have a small variation of < 5 % for $^{223}$Ra and $^{224}$Ra mass balance models.
The radium inputs from rivers and atmospheric deposits account for < 2 % of total inputs and represent a minor
uncertainty.





**Table 3.** Parameters and values used in water mass balance model and radium mass balance models.

| Parameters | Values and units |
|---|---|
| Constants | |
| Water area in the Bay ($S$) | $5.56\times10^8$ m² |
| Water volume in the Bay ($V$) | $4.39\times10^9$ m³ |
| Water mass balance model | |
| Precipitation ($P_T$) | $4.55\times10^6$ m³ d⁻¹ |
| Evapotranspiration ($E_T$) | $2.00\times10^6$ m³ d⁻¹ |
| Total river flux ($\sum_{i=1}^{m} Q_{r(i)}$) | $6.75\times10^5$ m³ d⁻¹ |
| SFGD flux ($Q_{SFGD}$) | $1.89\times10^6$ m³ d⁻¹ |
| ²²³Ra mass balance model | |
| ²²³Ra decay coefficient ($\lambda_{223}$) | 0.061 d⁻¹ |
| ²²³Ra inventory ($I_{223}$) | $1.36\times10^{11}$ dpm |
| $^{223}Ra_{gw\text{-}ns}$ | 21.6−68.8 dpm 100 L⁻¹ |
| ²²³Ra activity in the open sea water ($^{223}Ra_{op}$) | 1.14 dpm 100 L⁻¹ |
| ²²³Ra flux from atmospheric deposits ($^{223}F_{atm}$) | $7.96\times10^5$ dpm d⁻¹ |
| ²²³Ra flux from rivers ($^{223}F_r$) | $1.45\times10^7$ dpm d⁻¹ |
| ²²³Ra flux from sediments ($^{223}F_{sed}$) | $1.52\times10^9$ dpm d⁻¹ |
| SGD flux ($Q_{SGD}$) by ²²³Ra[a] | $(1.61−5.13)\times10^7$ m³ d⁻¹ |
| ²²⁴Ra mass balance model | |
| ²²⁴Ra decay coefficient ($\lambda_{224}$) | 0.189 d⁻¹ |
| ²²⁴Ra inventory ($I_{224}$) | $2.59\times10^{12}$ dpm |
| $^{224}Ra_{gw\text{-}ns}$ | 645−2133 dpm 100 L⁻¹ |
| ²²⁴Ra activity in the open sea water ($^{224}Ra_{op}$) | 24.10 dpm 100 L⁻¹ |





| | |
|---|---|
| $^{224}$Ra flux from atmospheric deposits ( $^{224}F_{atm}$ ) | $1.59 \times 10^7$ dpm d$^{-1}$ |
| $^{224}$Ra flux from rivers ( $^{224}F_r$ ) | $4.32 \times 10^8$ dpm d$^{-1}$ |
| $^{224}$Ra flux from sediments ( $^{224}F_{sed}$ ) | $9.86 \times 10^{10}$ dpm d$^{-1}$ |
| SGD flux ( $Q_{SGD}$ ) by $^{224}$Ra[a] | $(2.20-7.27) \times 10^7$ m$^3$ d$^{-1}$ |
| $^{226}$Ra mass balance model | |
| $^{226}$Ra decay coefficient ( $\lambda_{226}$ ) | $1.19 \times 10^{-6}$ d$^{-1}$ |
| $^{226}$Ra inventory ( $I_{226}$ ) | $6.81 \times 10^{11}$ dpm |
| $^{226}Ra_{gw-ns}$ | $14.9-47.9$ dpm 100 L$^{-1}$ |
| $^{226}$Ra activity in the open sea water ( $^{226}Ra_{op}$ ) | 12 dpm 100 L$^{-1}$ |
| $^{226}$Ra flux from atmospheric deposits ( $^{226}F_{atm}$ ) | $1.59 \times 10^7$ dpm d$^{-1}$ |
| $^{226}$Ra flux from rivers ( $^{226}F_r$ ) | $2.03 \times 10^8$ dpm d$^{-1}$ |
| $^{226}$Ra flux from sediments ( $^{226}F_{sed}$ ) | $1.26 \times 10^7$ dpm d$^{-1}$ |
| SGD flux ( $Q_{SGD}$ ) by $^{226}$Ra[a] | $(1.58-5.09) \times 10^7$ m$^3$ d$^{-1}$ |
| $^{228}$Ra mass balance model | |
| $^{228}$Ra decay coefficient ( $\lambda_{228}$ ) | $3.27 \times 10^{-4}$ d$^{-1}$ |
| $^{228}$Ra inventory ( $I_{228}$ ) | $1.42 \times 10^{12}$ dpm |
| $^{228}Ra_{gw-ns}$ | $45.0-77.9$ dpm 100 L$^{-1}$ |
| $^{228}$Ra activity in the open sea water ( $^{228}Ra_{op}$ ) | 17.67 dpm 100 L$^{-1}$ |
| $^{228}$Ra flux from atmospheric deposits ( $^{228}F_{atm}$ ) | $1.59 \times 10^7$ dpm d$^{-1}$ |
| $^{228}$Ra flux from rivers ( $^{228}F_r$ ) | $1.46 \times 10^8$ dpm d$^{-1}$ |
| $^{228}$Ra flux from sediments ( $^{228}F_{sed}$ ) | $2.76 \times 10^9$ dpm d$^{-1}$ |
| SGD flux ( $Q_{SGD}$ ) by $^{228}$Ra[a] | $(3.87-6.70) \times 10^7$ m$^3$ d$^{-1}$ |

[a]The range between the first to third quartiles of $^{22n}Ra_{gw-ns}$ is used to estimate the SGD.



### 4.5 Nutrient fluxes through SGD and primary production

Similar to radium quartet, nutrients in coastal waters include various sources such as SGD input, riverine input, diffusion from sediments, atmospheric deposits and mariculture. These external nutrient loadings will be consumed to support the growth of phytoplankton during primary production. Wang et al. (2018) constructed a DIP mass balance model and obtained a primary production of 54–73 mg C $m^{-2}$ $d^{-1}$ in Daya Bay. SGD has been regarded as an important pathway of nutrients fluxes from land to ocean, but the primary production supported by nutrient inputs through SGD is seldom reported in the bay.

Two recent studies evaluated the nutrient fluxes through SGD in winter and spring of Daya Bay by multiplying the SGD flux and nutrient concentrations of groundwater (Wang et al., 2018, Gao et al., 2018). However, their approach did not take into account the return nutrient fluxes that RSGD takes away from the sea and overestimated the nutrients via SGD. In this study, the nutrient fluxes from SGD $F_{SGD}$ are derived using the following equation:

$$F_{SGD} = Q_{SGD} N_{gw} - Q_{RSGD} N_{ns} \tag{11}$$

where $N_{gw}$ and $N_{ns}$ are the nutrient concentrations in groundwater and nearshore seawater, respectively. The mean values of nutrient concentrations are 68.3 µmol $L^{-1}$ for $NO_3$, 88.3 µmol $L^{-1}$ for DIN and 1.73 µmol $L^{-1}$ for DIP in all groundwater samples; and are 38.9 µmol $L^{-1}$ for $NO_3$, 55.8 µmol $L^{-1}$ for DIN and 1.14 µmol $L^{-1}$ for DIP in 8 nearshore seawater samples (S15–S19 and S22–S24). Based on the mean concentrations of nutrients, the SGD associated nutrient fluxes are estimated to be $(1.21–1.58)\times10^6$ mol $d^{-1}$ for $NO_3$, $(1.36–1.76)\times10^6$ mol $d^{-1}$ for DIN and $(2.53–3.26)\times10^4$ mol $d^{-1}$ for DIP in Daya Bay. Our result of $NO_3$ is approximately an order of magnitude greater than that derived from Wang et al. (2018). This difference between estimations may be a result of different methods and seasonal variations. If we assume that the nutrients in the region are consumed with the Redfield ratio, the DIN flux through SGD would support a primary production of 228–294 mg C $m^{-2}$ $d^{-1}$.

Riverine inputs have typically been considered as the major sources of nutrients to coastal water. Based on the nutrient concentrations in river water and the corresponding discharges of rivers, the DIN and DIP fluxes from four major rivers can be estimated to be $6.53\times10^4$ mol $d^{-1}$ and $1.81\times10^3$ mol $d^{-1}$, respectively. The DIN flux from rivers could provide





a primary production of 10.9 mg C m$^{-2}$ d$^{-1}$. For comparisons, the primary production supported by other external DIN is

also estimated in Daya Bay. Ni et al. (2017) estimated the diffusion rates (5.47×10$^{-4}$ mol m$^{-2}$ d$^{-1}$ for DIN and 2.25×10$^{-5}$ mol

m$^{-2}$ d$^{-1}$ for DIP) at the sediment-water interface. Multiplying these rates by the water area, we estimate the diffusion fluxes

in the whole bay to be 3.04×10$^{5}$ mol d$^{-1}$ for DIN and 1.25×10$^{4}$ mol d$^{-1}$ for DIP. The DIN flux from sedimentary input

sustains a primary production of 50.5 mg C m$^{-2}$ d$^{-1}$. Atmospheric deposition may be an important pathway for nutrients

entering aquatic ecosystems. Chen et al. (2014) evaluated the atmospheric dry and wet depositions of nutrients in different

seasons in Daya Bay. Based on their investigation, the total DIN and DIP deposits in summer are derived to be 1.97×10$^{5}$

mol d$^{-1}$ and 5.04×10$^{2}$ mol d$^{-1}$, respectively. Nitrogen and phosphorus can be discharged from aquaculture areas into coastal

waters. Recent research has shown that the inputs of DIN and DIP from mariculture in Daya Bay are 9.0×10$^{3}$ mol d$^{-1}$ and

5.30×10$^{2}$ mol d$^{-1}$, respectively (Wang et al., 2018). The primary production supported by DIN fluxes from atmospheric

deposits and mariculture is estimated to be 32.9 mg C m$^{-2}$ d$^{-1}$ and mg C m$^{-2}$ d$^{-1}$, respectively.

Figure 5a shows the DIN flux, DIP flux and their ratios from different sources (SGD, rivers, sediments, atmospheric

deposits and mariculture). Among all the sources of nutrients, SGD contributes up to 63.8 %–82.4 % and 57.1 %–73.6 %

of total inputs for DIN and DIP, respectively. The DIN and DIP fluxes through SGD exceed the inputs transported from the

local rivers by a factor of ~24 and 16, respectively. Despite the uncertainty of the nutrient concentrations in the

groundwater end-member and nearshore seawater end-member, we can conclude that SGD is a significant source of

nutrients into Daya Bay. The mean ratio of DIN/DIP in seawater is approximately 40.4. The depletion of inorganic

nitrogen over phosphorus in Daya Bay is indicative of a typical P-limited coastal system. On average, the ratio of the SGD

associated DIN flux to DIP flux is 54 and 3.4 times greater than the Redfield ratio (16). The high ratio suggests that the

nutrient inputs via SGD would result in the changes of nutrient structure and increase the risk of unbalance nutrient

condition in the seawater of Daya Bay.

Daya Bay is evaluated as a medium trophic level and eutrophication could be found in certain nearshore waters (Wu

and Wang, 2007). Primary production has been regarded as a good index of assessment of eutrophication status in coastal

waters and it is more sensitive on the response to aquatic environment (Liu et al., 2012). High primary production values

often occur in waters with frequent outburst of red tides, especially in spring and summer of Daya Bay (Wang et al., 2006).



The maximum primary production reaches up to 2722 mg C m$^{-2}$ d$^{-1}$ when a red tide appeared in Aotou Cove, located in the
northeast of Daya Bay (Song et al., 2004). In this study, the total primary production supported by all external DIN is
estimated to be 323–390 mg C m$^{-2}$ d$^{-1}$. By comparisons to other sources of DIN, this implies that the primary production
supported by DIN inputs from SGD accounts for about 73.1 % of total production (Fig. 5b). It is clear that the nutrient
fluxes through SGD contributes a significant fraction of primary production. Thus large inputs of nutrients from SGD may
be responsible for the increasing occurrences of harmful red tides in Daya Bay.
**5. Conclusions**
Seawater, groundwater and river water samples are collected for the analyses of radium quartet and nutrients based on a
field campaign conducted in July 2015 in Daya Bay, China. The SFGD, flushing time, SGD and nutrient fluxes through
SGD are evaluated with water and radium mass balance models. The primary production supported by all the external
nutrients are determined based on the Redfield ratio. The major conclusions are as follows.
(1) An improved flushing time model, which considers the effects of the five factors (rivers, open sea water
end-member, sedimentary input, atmospheric deposits and RSGD), evaluates a flushing time of 11.8–27.7 d in Daya Bay.
The analyses from seven different cases indicates that the open sea water end-member and RSGD have significant
influence on the flushing time estimation. In comparison with the results obtained from the improved model, previous
apparent age model overestimates the flushing time by 10.7 %–103 %.
(2) With the flushing time estimated by new model, the SGD flux of $(3.87–5.09)\times10^7$ m$^3$ d$^{-1}$ is derived from four
radium mass balance models. The SFGD flux is estimated to be $1.89\times10^6$ m$^3$ d$^{-1}$ based on a water mass balance model,
accounting for ~4.3 % of total SGD flux.
(3) Considering the return nutrient fluxes, the DIN and DIP fluxes through SGD are estimated to be $(1.36–1.76)\times10^6$
mol d$^{-1}$ and $(2.53–3.26)\times10^4$ mol d$^{-1}$, respectively. Among all the sources of nutrients, one can find that SGD is the
predominant source, contributing up to 63.8 %–82.4 % and 57.1 %–73.6 % of total inputs for DIN and DIP, respectively.
Large amounts of nutrient inputs from SGD and high N/P ratio in SGD may have more important influences on the
structure and balance of nutrients.



(4) The DIN flux through SGD would support a primary production of 228–294 mg C $m^{-2}$ $d^{-1}$, accounting for about
73.1 % of total primary production. By comparison to other external DIN, SGD supports a significant fraction of primary
production. The results show that the nutrient fluxes carried by SGD may be responsible for the frequent outbreaks of
harmful red tides in the coastal zone. This study may provide useful information for the management of coastal ecological
environment.

*Data availability.* Precipitation data are from National Meteorological Information Center (NMIC) at http://data.cma.cn/.
River discharge data are from Ren et al. (2013). Radium and nutrient data can be found in supplementary information and
they are available upon request by contacting the correspondence author.

*Competing interests.* The authors declare that they have no conflict of interest.

*Acknowledgements.* This work was supported by the National Basic Research Program of China ("973" Program, Grant
Nos. 2015CB452902 and 2015CB452901), the National Natural Science Foundation of China (Grant No. 41272267), and
the Guangdong Provincial Key Laboratory of Soil and Groundwater Pollution Control (No. 2017B030301012). The
authors thank Yanman Li, An An, Zongzhong Song, and Shaohong Li for their field work.

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





**Figures**

**Figure 1.** The study area and sampling stations in Daya Bay, China. The dots, diamonds and triangles denote sampling stations of

seawater, groundwater and river water, respectively.





**Figure 2.** Spatial distributions of $^{223}$Ra, $^{224}$Ra, $^{226}$Ra and $^{228}$Ra activities (dpm 100 L$^{-1}$) in surface and bottom seawater of Daya Bay in

July 2015.





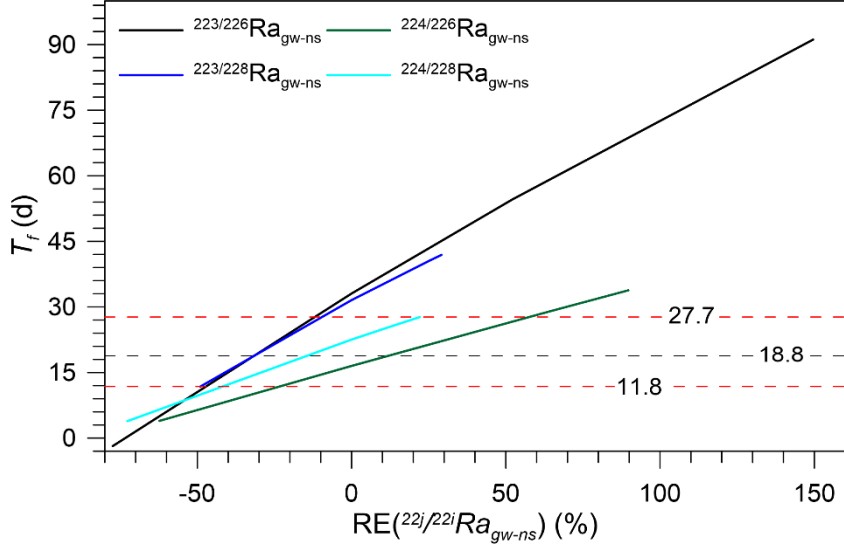

**Figure 3.** Changes of flushing time $T_f$ for four different combinations of radium quartet with $RE(^{22j/22i}Ra_{gw-ns})$. Four solid

lines of different colors denote the $T_f$ values estimated by four ratios ($^{223/226}Ra_{gw-ns}$, $^{223/228}Ra_{gw-ns}$, $^{224/226}Ra_{gw-ns}$ and

$^{224/228}Ra_{gw-ns}$), when $^{22j/22i}Ra_{gw-ns}$ ranges from $Q_1(^{22j/22i}Ra_{gw-ns})$ to $Q_3(^{22j/22i}Ra_{gw-ns})$. The horizontal red dashed

lines denote their common intersection and the black dashed line denotes the result of tidal prism model.





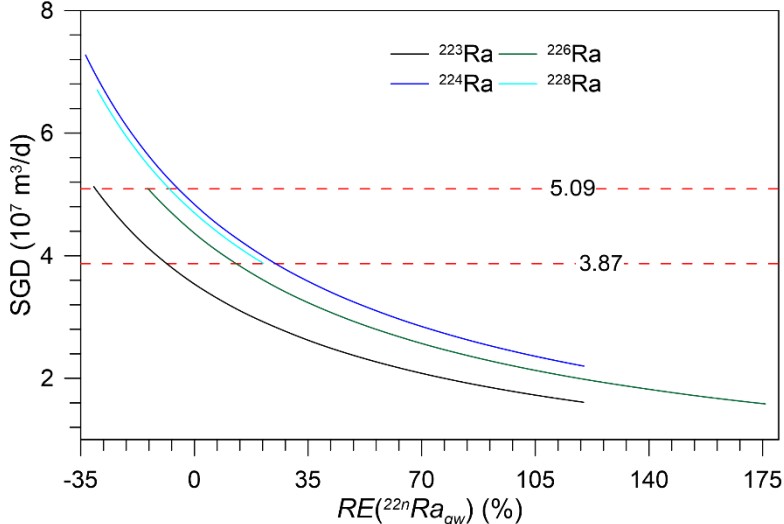


**Figure 4.** Changes of SGD estimated by $^{223}$Ra, $^{224}$Ra, $^{226}$Ra and $^{228}$Ra mass balance models with $RE(^{22n}Ra_{gw})$ when

$^{22n}Ra_{gw} = Q_1(^{22n}Ra_{gw}) \sim Q_3(^{22n}Ra_{gw})$. The red dashed lines denote their common intersection.





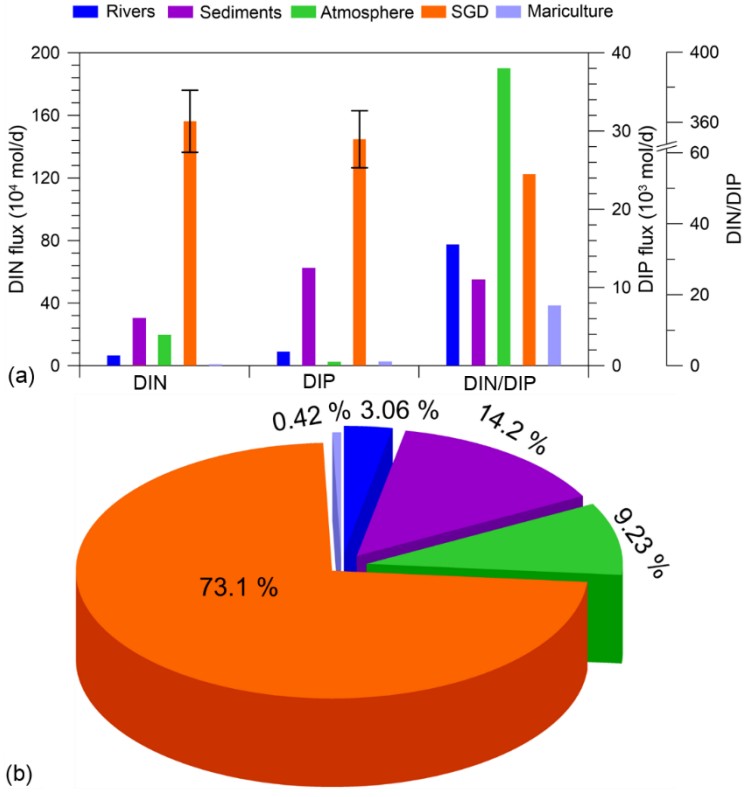


**Figure 5.** Comparisons of (a) nutrient fluxes, and (b) proportion of primary production supported by five different sources (SGD, rivers,

sediments, atmospheric deposits and mariculture) in summer of Daya Bay.