# Peer review of "Evaluation of flushing time, groundwater discharge and associated 1 nutrient fluxes in Daya Bay, China 2 3 Yan Zhang1,2,3†, Meng Zhang4†, Hailong Li2,3\*, Xuejing Wang3, Wenjing Qu1,2, Xin Luo5,6, Kai Xiao1,2, 4 Xiaolang Zhang3 5 6 7 1MOE Key Laboratory of Groundwater Circulation and Environment Evolution and School of Water Resources and 8 Environment, China University of Geosciences-Beijing, Bei"

_Hydrology and Earth System Sciences, 2018_

## Referee Comment (RC1) · Anonymous Referee #1 · 25 May 2018

The manuscript entitled "Evaluation of flushing time, groundwater discharge and associated fluxes in Daya Bay, China" presents the application of the different isotopes of radium for establishing an improved water budget of Daya bay in China. The authors calculated additional sources and sinks or radium based on several assumptions and partially new data to obtain a more complete overview of fluxes in the study area later applied for estimating the nutrient delivery to the bay. Daya Bay presents serious contamination issues connected with these processes and therefore it might be an interesting case study.

General comments:

[Figure]

I have doubts about the novelty of this work since it has been published a similar article recently (Wang et al., 2018) . As an example, the tittle of the work published and this one are very similar indicating almost exactly the same content. I detail these concerns below.

The abstract and introduction point towards two objectives: (1) a global objective referring to models neglecting the effect of rivers, open sea water end member, sedimentary input atmospheric deposit and recirculated seawater and (2) a local objective associated with the improvement of the water budget and the delivery of nutrients to Daya Bay. Nevertheless, I think that the background information provided in both cases in not sufficient to show the relevance of the topics from a global perspective. If the manuscript pretends to show a progress in the application of the methodology, this should be described and presented in the introduction with references to previous studies applying these methods. The methodology applied is not something new and it would be needed to have a better introduction about what is the new contribution of this work. In principle, it seems that the method followed is just a replica of Zhang et al. (2017) in another bay. If the study is oriented to be a local improvement of the water budget in the studied region, previous studies in the area should be clearly presented and a discussion about what can be improved on them. There are data already published in this work that therefore should be properly cited as for example the 224Ra distribution, salinity distribution or the nutrients information with very similar figures and graphs. New and previously published information should be clearly defined to evaluate the novel contribution of this work.

The explanations about the return nutrient fluxes that RSGD takes away from the sea and the overestimation of the nutrients via SGD should be better explained. A simple mention to previous published papers is not enough to understand the problem that requires to be solved. The authors should document better what are the reasons and processes that lead to them to think in this way.

Considering that nutrients samples vary 2 orders of magnitude (3-300 for NO3), how

reliable can be considered the estimations based on "the mean concentrations of nutrients"?. For example, if I eliminate sample GW2, the arithmetic mean would be around half and that would approximate the results to the previous estimates of Wang et al (2018). Is the objective of the manuscript to provide a much better estimate of this value and not a similar one?.

Since one of the main objectives of the manuscript is to improve the knowledge about the water fluxes and nutrients fluxes in Daya Bay, a full comparison and discussion with previous estimates should be presented to have a more quantitative overview of the improvement acquired with this study. This would be also useful to evaluate the interest of this research as a case study and if enough novel content is presented to be published.

The differences in flushing time for the 7 cases presented are really small for the first 4 cases and it seems that it is only relevant for the connection with the sea and the RSGD, still along the text it is mentioned the impact of the rivers, sediments and atmospheric deposits as key elements for the Ra budget of the area. Seeing the results, this can sound inconsistent and especially since most of these calculations are based on generalizations/assumptions and not based on new collection of data (hence there is not new information other than the calculations).

Lines 181-188. The comparison between SFGD of different coastal systems using the discharge divided by the total area of the bay does not have a physical sense since the freshwater discharge is not taking place at distant locations from the shore. Also the climatic conditions (rain, evaporation) and the size of the catchment on land would be what would play a major role on these numbers, therefore I think that the matching of values is accidental and shouldn't be used as a reference of the quality of the calculations.

---

## Author Comment (AC1) · 15 Jun 2018

Response to Interactive Comment by Anonymous Referee #1

The manuscript entitled "Evaluation of flushing time, groundwater discharge and associated fluxes in Daya Bay, China" presents the application of the different isotopes of radium for establishing an improved water budget of Daya bay in China. The authors calculated additional sources and sinks or radium based on several assumptions and partially new data to obtain a more complete overview of fluxes in the study area later applied for estimating the nutrient delivery to the bay. Daya Bay presents serious contamination issues connected with these processes and therefore it might be an

interesting case study.

Many thanks for your recognition of our work and its significance.

General comments:

1. I have doubts about the novelty of this work since it has been published a similar article recently (Wang et al., 2018). As an example, the tittle of the work published and this one are very similar indicating almost exactly the same content. I detail these concerns below.

Response 1:

The newly published paper by Wang et al. (2018) entitled "Submarine groundwater discharge as an important nutrient source influencing nutrient structure in coastal water of Daya Bay, China", evaluated SGD and associated nutrient fluxes based on the 224Ra and nutrient data collected in December 2015 in Daya Bay. Our current study is titled "Evaluation of flushing time, groundwater discharge and associated fluxes in Daya Bay, China". To distinguish the content of this research from that of Wang et al. (2018), in the future revision we will change the original title into "Improvement of evaluation of flushing time and submarine groundwater discharge: a case study in Daya Bay, China". Correspondingly, we will emphasize the importance of the improved flushing time model and adjust the structure of the current study based on the following three major sections: (1) Introduction, (2) Improved flushing time model and (3) Application in Daya Bay, China.

In fact, there are significant differences between Wang et al. (2018) and the current study, which can be summarized as follows:

First, the sampling time for the two study is different. Field campaigns were conducted in the wet season (July 2015) for our study, but in the dry season (December 2015) for the previous study by Wang et al. (2018). The daily precipitation and riverine discharge in the wet season are approximately 5 and 2 times greater than those in the dry one,

[Figure]

respectively (Supplementary Table 1). Thus, the two studies reveal the variations in SGD and associated nutrient fluxes under two different hydrologic conditions.

Secondly, Wang et al. (2018) obtained an approximated flushing time using tidal prism model. The highlight in our research lies in the accurate estimation of flushing time by developing improved model which considers the effects of rivers, open sea water end-member, sedimentary input, atmospheric deposits and recirculated seawater (RSGD). Among the five factors, open sea water end-member, sedimentary input, and RSGD have significant effects on the results of flushing time of Daya Bay, but they are often neglected in many previous research.

Thirdly, the method for estimating the SGD and associated nutrient fluxes reported by Wang et al. (2018) did not take into account the returned flux that RSGD takes away from seawater. Moreover, the SGD estimated by Wang et al. (2018) is based on a 224Ra mass balance model, but the SGD in our paper is estimated by making full use of all the radium quartet (223Ra, 224Ra, 226Ra and 228Ra) data.

Finally, Wang et al. (2018) assessed the contributions of nutrients supported by different sources (SGD, benthic sediments, local rivers and atmospheric deposition) and showed that SGD is a key source of nutrients influencing nutrient structure in coastal waters. Our research, however, focused on the comparisons of primary production supported by various sources (see section 4.5). In addition, we also discussed the relationship between primary production supported by DIN inputs via SGD and harmful red tides, and confirmed the importance of the primary production attributed to SGD. In short, there are essential differences between the two studies.

2. The abstract and introduction point towards two objectives: (1) a global objective referring to models neglecting the effect of rivers, open sea water end member, sedimentary input atmospheric deposit and recirculated seawater and (2) a local objective associated with the improvement of the water budget and the delivery of nutrients to Daya Bay. Nevertheless, I think that the background information provided in both

cases in not sufficient to show the relevance of the topics from a global perspective. If the manuscript pretends to show a progress in the application of the methodology, this should be described and presented in the introduction with references to previous studies applying these methods. The methodology applied is not something new and it would be needed to have a better introduction about what is the new contribution of this work. In principle, it seems that the method followed is just a replica of Zhang et al. (2017) in another bay. If the study is oriented to be a local improvement of the water budget in the studied region, previous studies in the area should be clearly presented and a discussion about what can be improved on them. There are data already published in this work that therefore should be properly cited as for example the 224Ra distribution, salinity distribution or the nutrients information with very similar figures and graphs. New and previously published information should be clearly defined to evaluate the novel contribution of this work.

Response 2:

Compared to previous apparent age model, our improved flushing time model takes into account the effects of five factors, i.e., radium inputs from rivers, sediments and atmospheric deposits, open sea water end-member and RSGD. Thus, the objectives of our study are to (1) develop an improved flushing time model which could be applied in Daya Bay and other coastal systems; and (2) to estimate the SGD of Daya Bay based on the flushing time by new model and the method reported by Zhang et al. (2017); and (3) to assess the primary production supported by DIN inputs from SGD and its influence on ecological environment in Daya Bay.

In general, the manuscript aims to show a progress in the application of the methodology. According to referee's comments, we will add more descriptions about the existing models in previous studies and the contribution of the current work in Introduction Section in the later revision:

"Moore (2000) and Moore et al. (2006) developed the apparent age model based on

mass balance of radium quartet, which has been widely used to quantify the flushing time in many previous studies (e.g., Peterson et al., 2008; Ji et al., 2013; Tomasky-Holmes et al., 2013; Xu et al., 2013; Wang et al., 2015; Luo and Jiao, 2016). However, the model assumed that groundwater is the major source, and neglected other sources such as riverine input, sedimentary input and atmospheric deposits. Moreover, it did not consider the effects of open sea water end-member and RSGD. Thus, the flushing time by apparent age model may be of considerable uncertainties, but the uncertainties induced by the above-mentioned five factors are seldom discussed. Here we develop an improved flushing time model which includes all sources and sinks to enhance the accuracy of flushing time by coupling two radium mass balance models. The improved model could be applied in Daya Bay and other coastal systems elsewhere."

In fact, Zhang et al. (2017) presented an improved method which considers the losses of tracers caused by RSGD to enhance accuracy in estimating SGD and they mainly discussed the influence of RSGD on the SGD for tracer-based models based on theoretical and data analyses. Our current research focuses on the accurate estimation of flushing time by developing improved flushing time model and mainly discusses the influence of various factors including RSGD on the flushing time.

It should be noted that all data used in this manuscript are never published in other journals. I am afraid that the referee just has a misunderstanding about the data used in Wang et al. (2018) and our research (see Response 1). In order to eliminate the misunderstanding, in the revised manuscript we will highlight the comparisons with previous study in this area (Wang et al. 2018) and the novel contribution of our current research (see Response 1).

3. The explanations about the return nutrient fluxes that RSGD takes away from the sea and the overestimation of the nutrients via SGD should be better explained. A simple mention to previous published papers is not enough to understand the problem that requires to be solved. The authors should document better what are the reasons and processes that lead to them to think in this way.

Response 3:

Thanks for this suggestion. In the revision, we will add the following explanations about the return nutrient fluxes that RSGD takes away from the sea and the overestimation of the nutrients via SGD:

Driven by both terrestrial and marine forcing components, SGD comprises SFGD and RSGD. RSGD results from the intrusion of seawater that invades coastal aquifers and subsequently flows back into the ocean. Thus RSGD does not only deliver 'new nutrients' into the sea when it flows into the sea from the aquifer but also takes away 'old nutrients' from seawater system when it enters coastal aquifers. However, the concentrations of 'new nutrients' and 'old nutrients' are definitely different because of chemical and biological processes in the mixing zone. Although RSGD does not affect the water balances of the aquifers and surface water, it modifies significantly tracer and nutrient balances.

The most common method for quantifying nutrient fluxes via SGD is to multiply the SGD flux by their concentrations in groundwater. The approach did not take into account the return nutrient fluxes that RSGD takes away from the sea and therefore overestimated the nutrients via SGD. More details about the overestimation of nutrient fluxes via SGD are shown in Response 4.

4. Considering that nutrients samples vary 2 orders of magnitude (3-300 for NO3), how reliable can be considered the estimations based on "the mean concentrations of nutrients"? For example, if I eliminate sample GW2, the arithmetic mean would be around half and that would approximate the results to the previous estimates of Wang et al (2018). Is the objective of the manuscript to provide a much better estimate of this value and not a similar one? Since one of the main objectives of the manuscript is to improve the knowledge about the water fluxes and nutrients fluxes in Daya Bay, a full comparison and discussion with previous estimates should be presented to have a more quantitative overview of the improvement acquired with this study. This would

be also useful to evaluate the interest of this research as a case study and if enough novel content is presented to be published.

Response 4:

Yes. If excluding the abnormally high value observed at GW2ïïjŇthe arithmetic mean of 33.8 $\mu$mol/L for groundwater NO3 would be around half of the original value (68.3 $\mu$mol/L) used in our paper and it is approximated by the highest value (31.43 $\mu$mol/L) obtained from Wang et al (2018). The objective is not to compare our result with that of Wang et al. (2018) because the two studies were conducted in different seasons and revealed the seasonal variations (see Response 1).

As mentioned by referee, the NO3 concentrations in groundwater are highly variable due to heterogeneity of coastal aquifers, with a high standard deviation of 102.17 $\mu$mol/L. If the first quartile or median of groundwater NO3 data is used, the negative NO3 flux via SGD would be derived. Therefore, determining an exact groundwater end-member concentration in this study would still be a difficult challenge. To obtain reasonable groundwater end-member values, the more groundwater samples will be collected for nutrient analyses in July 2018. In this revision, three different nutrient end-members (mean concentration, mean concentration $\pm$ standard deviation for all groundwater nutrient samples collected in July 2015 and July 2018) will be used to assess the uncertainty induced by the end-member selection.

According to the suggestion of referee, a full comparison and discussion with previous estimates about the water fluxes and nutrients fluxes will be added in the revision:

1) About the water fluxes (SGD)

Zhang et al. (2017) developed a new model to assess the SGD of coastal waters. Compared with previous existing tracer-based models, the new model for SGD estimation considered the tracer losses carried by RSGD from seawater. The comparisons of the SGD by new model and existing model in JZB and previous studies were made in

Zhang et al. (2017). Based on their theoretical analysis, neglecting the losses of tracers induced by RSGD would underestimate the SGD by a percentage approximately equaling the tracer activity ratio of nearshore seawater to groundwater (AR). To eliminate the underestimation of SGD, the new model is applied in Daya Bay, China. The ranges between the first and third quartiles of AR in Daya Bay are 0.06~0.19 for 223Ra, 0.05~0.17 for 224Ra, 0.29~0.61 for 226Ra, and 0.29~0.54 for 228Ra. Using the ratio of SFGD to SGD (4.3 %), one can find that the existing old model underestimated the SGD by 5.36~18.22 % for 223Ra model, 4.67~15.96 % for 224Ra model, 28.5~60.2 % for 226Ra model and 28.3~53.2 % for 228Ra model in Daya Bay. The larger the activity ratio of tracers AR is, the higher the underestimation of SGD is, which is in accordance with previous study (Zhang et al. 2017).

2) About the nutrient fluxes via SGD

In previous studies, the nutrient fluxes via SGD FOSGD are derived using the following equation:

$$FOSGD = QSGD*Ngw \quad (1)$$

In Daya Bay, the nutrient fluxes from SGD FNSGD are derived using the following equation:

$$FNSGD = QSGD*Ngw - QRSGD*Nns \quad (2)$$

where Ngw and Nns are the nutrient concentrations in groundwater and nearshore seawater, respectively. The new method (Eq. (2)) considers the return nutrient fluxes that RSGD takes away from seawater. In order to quantify such improvement, we define the relative error (REN) as

$$REN = (FOSGD - FNSGD)/FNSGD \quad (3)$$

Substituting equations (1) and (2) into (3), yields

$$REN = (1-RF)*NR/[1-(1-RF)*NR] \quad (4)$$

where RF is the ratio of SFGD to SGD; NR is the nutrient concentration of nearshore seawater to groundwater.

In Daya Bay, the mean NR is 0.57 for NO3, 0.63 for DIN and 0.66 for DIP. With Eq. (4), one can find that the the nutrient fluxes via SGD in previous estimates (Eq. (1)) are overestimated by 152 %, 168 % and 120 % for NO3, DIN and DIP, respectively.

5. The differences in flushing time for the 7 cases presented are really small for the first 4 cases and it seems that it is only relevant for the connection with the sea and the RSGD, still along the text it is mentioned the impact of the rivers, sediments and atmospheric deposits as key elements for the Ra budget of the area. Seeing the results, this can sound inconsistent and especially since most of these calculations are based on generalizations/assumptions and not based on new collection of data (hence there is not new information other than the calculations).

Response 5:

Indeed, the differences in flushing time are really small for the first 4 cases. The results from 7 cases show that the flushing time is strongly influenced by the open sea water end-member and RSGD in Daya Bay, while other factors (especially for rivers and atmospheric deposits) have minor effect on the flushing time estimation. We have the following explanations about the impact of the rivers, sediments and atmospheric deposits:

In general, radium input from atmospheric deposit is ignorable in most coastal environments due to low activities. Thus the atmospheric deposit indeed has minor effect on the result of flushing time for Daya Bay and many other coastal systems.

Sedimentary input is not ignorable for most radium quartet in an embayment except for 226Ra due to its low production rate in marine sediments and long half-life. In Daya Bay (the current study), one can see that neglecting the radium inputs from sediments would underestimate the flushing time by ∼20 % with the ratio of radium quartet

(224Ra/228Ra). Thus sedimentary input has an important effect on the result of flushing time.

Riverine input in Daya Bay contributes a quite small portion (<2%) and could be neglected. However, riverine input is not ignorable in many other coastal systems, especially for river-dominated areas such as Laizhou Bay, China. The Yellow River, as the sixth largest river in the world, is the largest that discharges into the Laizhou Bay. Wang et al. (2015) estimated a flushing time of 31.3∼41.9 d in Laizhou Bay based on an apparent age model with respect to 223Ra and 226Ra which neglects the effects of Yellow River and other rivers. If considering the riverine inputs of 223Ra and 226Ra, the flushing time of 23.6∼29.6 d in Laizhou Bay will be derived. It can be seen that ignoring the riverine inputs of radium results in significant increases (32.6∼41.6 %) in flushing time in Laizhou Bay.

It cannot be concluded that riverine input has minor effect on the flushing time estimation for any other coastal systems, but only for Daya Bay. Based the above discussion, significant errors may be produced for trace-derived flushing time if one do not consider the effects of rivers, sedimentary input, RSGD and the open seawater end-member. Therefore, in order to obtain accurate result of flushing time, we recommend the improved model to assess the flushing time of coastal waters. The above discussion will be summarized in the later revision.

The referee suggests that our calculations are not based on new collection of data. In fact, only radium desorption from atmospheric deposit and riverine SPM is based on previous estimates. Moreover, the desorption flux contributes a quite small portion. We make sure that the radium and nutrient data for all seawater, groundwater and river water samples used in this paper indeed are new data and have never published in other journals.

6. Lines 181-188. The comparison between SFGD of different coastal systems using the discharge divided by the total area of the bay does not have a physical sense

since the freshwater discharge is not taking place at distant locations from the shore. Also the climatic conditions (rain, evaporation) and the size of the catchment on land would be what would play a major role on these numbers, therefore I think that the matching of values is accidental and shouldn't be used as a reference of the quality of the calculations.

Response 6:

Yes, you are right. Indeed, SFGD of different coastal systems is controlled by various factors such as climatic conditions and the size of the catchment. The simple comparison of values does not have a physical sense. Thus, we will delete the section (Lines 181-188) in the later revision.

References:

Moore, W.S., 2000. Ages of continental shelf waters determined from 223Ra and 224Ra. Journal of Geophysical Research: Oceans, 105(C9): 22117-22122. DOI:10.1029/1999jc000289.

Moore, W. S., Blanton, J. O., & Joye, S. B. (2006). Estimates of flushing times, submarine groundwater discharge, and nutrient fluxes to Okatee Estuary, South Carolina. Journal of Geophysical Research, 111(C9). doi: 10.1029/2005jc003041.

Peterson, R. N., Burnett, W. C., Taniguchi, M., Chen, J., Santos, I. R., & Misra, S. (2008). Determination of transport rates in the Yellow River–Bohai Sea mixing zone via natural geochemical tracers. Continental Shelf Research, 28(19), 2700-2707. doi: 10.1016/j.csr.2008.09.002.

Wang, X., Li, H., Jiao, J. J., Barry, D. A., Li, L., Luo, X., Qu, W. (2015). Submarine fresh groundwater discharge into Laizhou Bay comparable to the Yellow River flux. Sci Rep, 5, 8814. doi: 10.1038/srep08814.

Wang, X., Li, H., Zheng, C. M., Yang, J. Z., Zhang, Y., Zhang, M., Qi, Z. H., Xiao, K., Zhang, X. (2018). Submarine groundwater discharge as an important nutrient source

influencing nutrient structure in coastal water of Daya Bay, China. Geochimica et Cosmochimica Acta 225 (2018) 52-65.

Zhang, Y., Li, H., Xiao, K., Wang, X., Lu, X., Zhang, M., An, A., Qu, W., Wan, L., Zheng, C., Wang, X., Jiang, X. (2017). Improving Estimation of Submarine Groundwater Discharge Using Radium and Radon Tracers: Application in Jiaozhou Bay, China. Journal of Geophysical Research: Oceans. DOI:10.1002/2017jc013237.

Luo, X., & Jiao, J. J. (2016). Submarine groundwater discharge and nutrient loadings in Tolo Harbor, Hong Kong using multiple geotracer-based models, and their implications of red tide outbreaks. Water Res, 102, 11-31. doi: 10.1016/j.watres.2016.06.017.

Ji, T., Du, J., Moore, W. S., Zhang, G., Su, N., & Zhang, J. (2013). Nutrient inputs to a Lagoon through submarine groundwater discharge: The case of Laoye Lagoon, Hainan, China. Journal of Marine Systems, 111-112, 253-262. doi: 10.1016/j.jmarsys.2012.11.007.

Xu, B., Burnett, W., Dimova, N., Diao, S., Mi, T., Jiang, X., & Yu, Z. (2013). Hydrodynamics in the Yellow River Estuary via radium isotopes: Ecological perspectives. Continental Shelf Research, 66, 19-28. doi: 10.1016/j.csr.2013.06.018.

Tomasky-Holmes, G., Valiela, I., & Charette, M. A. (2013). Determination of water mass ages using radium isotopes as tracers: Implications for phytoplankton dynamics in estuaries. Marine Chemistry, 156, 18-26. doi: 10.1016/j.marchem.2013.02.002.

Please also note the supplement to this comment:
https://www.hydrol-earth-syst-sci-discuss.net/hess-2018-207/hess-2018-207-AC1-supplement.pdf

---

## Referee Comment (RC2) · Anonymous Referee #2 · 6 Jul 2018

My main concern about this paper is the methodology. The sampling is very located in time (28 to 31 july 2015). The measured concentration of Radium shows high spatial variability (p. 7), the same for nutrients. This shows that the system is very dynamic. However, the model used for flux balance is in steady state. Taking care of the highly variable meteorological conditions, the sampled concentrations are not representative of steady state conditions. Therefore, the estimated water and solute fluxes are highly questionable.

Furthermore, the manuscript lacks some information like: - Is evapotranspiration the potential one? (should not be). How is the actual evapotranspiration estimated? - The

tidal prism is depending on the GW level. Please explain in details how it is taken into account. - Assumptions related to eq. 7 should be clearly stated. . .

---

## Author Comment (AC2) · 12 Jul 2018

In this study, we estimated water and nutrient fluxes based on the steady state models that neglect the variation of radium storage in the bay. The referee doubts the validity of steady state models. About the steady state in our model, we have the following explanations:

1. This steady state is a common and typical approach for tracer-based SGD studies and has been used in all the previous studies of SGD by radium isotope methods (e.g., Moore, 1996; Kim et al. 2005; Moore et al., 2006; Moore et al., 2008).

2. During our observation period (wet season), both SFGD and rivers approximately reach their maximum discharging rates so that the mass of radium in the whole bay is approximately at its maximum. In other words, the observation time is near an extreme point of the function of total radium mass in the bay with respect to time. Since the variation of radium storage in the whole bay approaches zero near the extreme point, the steady state is approximately valid during our sampling period.

3. The referee suggests that the bay system is very dynamic due to the high spatial variability of radium. In fact, the radium concentrations in seawater are associated with various factors, such as offshore distance, lithology and geological conditions. From point scale, the radium concentrations vary significantly among stations. When taking the whole bay as the study subject, the system is under steady-state condition (see explanation 2). The SGD and nutrient fluxes in our study were estimated based on the scale of the whole bay. Moreover, the meteorological conditions were stable and did not show obvious variations during our sampling period. Thus the sampled concentrations could represent the steady state conditions and the steady state used in this study is reasonable.

Strict quantification of the error induced by steady state assumption needs not only much more radium measurement data in terms of time series, but also quantification of the seawater flow in the whole bay for a long period (at least one year) by a numerical model. This will be a major task and is indeed beyond the scope of this paper.

The estimated evapotranspiration is actual one in our study. The actual evapotranspiration in this basin was estimated as the product of the area of the basin and daily evapotranspiration. The daily evapotranspiration was assessed based on the monthly evapotranspiration. The monthly evapotranspiration was estimated by the Gaoqiao equation (Guo and Li, 2008) using monthly mean air temperature and precipitation data from 2005 to 2015.

The referee indicates that tidal prism is depending on the GW level. In fact, the tidal

prism is associated with sea level (not GW level). Because the intertidal zone in Daya Bay is narrow, the change of the water surface area over the tidal cycle can be neglected. Thus, the tidal prism can be estimated approximately as the product of the tidal range and water surface area.

With the dispersion coefficient, sediment mixing coefficient, adsorption coefficient, production rate of the radionuclide in sediments and porosity, the radium input from sediments could be estimated based on an empirical formula (Eq. (7)) reported by Moore et al. (2011). In Eq. (7), the values of the dispersion coefficient and sediment mixing coefficient for typical marine sediments could be found in Zlotnik et al. (2010) and Moore et al. (2011). The activities of radium quartet in sediments can be obtained based on the contents of uranium (U) and thorium (Th) in coastal sediments of Guangdong Province, China. Using the radium activities in sediments, leachable ratios and sediment density, we can estimate the production rate of radium quartet. The calculation details for each term in Eq. (7) see Section 4.3.3. No assumptions for Eq. (7) were used.

References

Moore, W. S. (1996). Large groundwater inputs to coastal waters revealed by 226Ra enrichments. Nature, 380(6575), 612-614.

Moore, W. S., Blanton, J. O., & Joye, S. B. (2006). Estimates of flushing times, submarine groundwater discharge, and nutrient fluxes to Okatee Estuary, South Carolina. Journal of Geophysical Research, 111(C9). doi: 10.1029/2005jc003041.

Moore, W. S., Sarmiento, J. L., & Key, R. M. (2008). Submarine groundwater discharge revealed by 228Ra distribution in the upper Atlantic Ocean. Nature Geoscience, 1(5), 309-311. doi: 10.1038/ngeo183.

Moore, W.S., Beck, M., Riedel, T., Rutgers van der Loeff, M., Dellwig, O., Shaw, T.J., Schnetger, B., Brumsack, H.J. (2011). Radium-based pore water fluxes of silica, alkalinity, manganese, DOC, and uranium: A decade of studies in the German Wadden Sea. Geochimica et Cosmochimica Acta, 75(21), 6535-6555. DOI:10.1016/j.gca.20.

Zlotnik, V.A., Robinson, N.I., Simmons, C.T. (2010). Salinity dynamics of discharge lakes in dune environments: conceptual model. Water Resour. Res. 46 (11).

Guo, J., Li, G.P. (2008). Multi-time scales analysis of variations of water resource in Mount Emei from 1951 to 2005. Scientia Meteorologica Sinica, 5, 552-557.

Kim G., Ryu J.-W., Yang H.S., Yun S.T. (2005). Submarine groundwater discharge (SGD) into the Yellow Sea revealed by 228Ra and 226Ra isotopes: implications for global silicate fluxes. Earth Planet. Sci. Lett. 237, 156-166.